# Microstructural Study of CrNiCoFeMn High Entropy Alloy Obtained by Selective Laser Melting

**DOI:** 10.3390/ma15165544

**Published:** 2022-08-12

**Authors:** Enrico Gianfranco Campari, Angelo Casagrande

**Affiliations:** 1Department of Physics and Astronomy, Alma Mater Studiorum-University of Bologna, Viale Berti Pichat 6/2, 40127 Bologna, Italy; 2Department of Industrial Engineering, Alma Mater Studiorum-University of Bologna, Viale Risorgimento 4, 40136 Bologna, Italy

**Keywords:** high entropy alloys, selective laser melting, microstructure, mechanical properties, strengthening mechanism

## Abstract

The high entropy alloy (HEA) of equiatomic composition CrNiFeCoMn and with FCC crystal structure was additively manufactured in a selective laser melting (SLM) process starting from mechanically alloyed powders. The as-produced alloy shows fine nitride and σ phase precipitates, which are Cr-rich and stable up to about 900 K. The precipitates increase in number and dimensions after long-period annealing at 900–1300 K, with a change in the HEA mechanical properties. Higher aging temperatures in the furnace, above 1300 K, turn the alloy into a single FCC structure, with the disappearance of the nitride and σ phase precipitates inside the grains and at the grain boundaries, but still with the presence of a finer Cr-rich nitride precipitation phase. These results suggest that the as-produced HEA is a supersaturated solid solution at low and intermediate temperature with nitrides and σ nanostructures.

## 1. Introduction

The concept of high entropy alloy (HEA) was proposed and published in 2004 [1,2,3] and the unique properties of these alloys have attracted attention from research groups all over the world. The pioneering studies by Cantor et al. were the first to report that an equiatomic alloy consisting of the five transition metals Cr, Mn, Fe, Co and Ni crystallized as a single solid solution [1,4] in the as-cast and in the homogenized state.

The first HEA generation could contain five or more main elements with the concentration of each element in the range 5–35 at. % [3]. The alloys presented a high mixing entropy in their liquid state [4] and thus could form simple solid solution structures (e.g., FCC, BCC and HCP) in place of intermetallic phases. Some promising technological characteristics of HEAs were high hardness [5], good wear resistance [6], excellent strength at both high and low temperatures [7,8,9] and generally a good resistance to oxidation and corrosion [10]. The unique properties of the HEAs were ascribed to the inherent properties of multicomponent solid solution formation, such as the distorted lattice structures [11], cocktail effect [12], sluggish diffusion [5] and formation of nanoscale deformation twins [8].

From a metallurgical standpoint, the suppression of the brittle intermetallic phases in such alloys is to be regarded as a really interesting feature. As recalled by other authors, the CrNiMnCoFe HEAs are considered stable disordered supersaturated FCC solid solutions with high ductility and remarkable fracture toughness [9]. HEAs usually exhibit good thermal stability [7,8,13], which is commonly attributed to sluggish long-range substitutional diffusion because of the lack of a major diffusion element and the need for cooperative diffusion of constituent atoms in order to have proper composition partitioning. Consequently, diffusion-related processes, including crystallization and grain growth, are expected to be slow in HEAs. 

The belief that the equiatomic CoCrFeNiMn alloy is an HEA example with a single disordered solid solution structure was recently challenged by the discovery of second phases in the alloy after low-temperature annealing. Despite the use of specific processing conditions (prolonged annealing or severe plastic deformation prior to annealing) these studies, contrasting with the previous literature, showed that a second phase formation can occur in this alloy because the sluggish diffusion in HEAs is not a guarantee of a stable structure and precipitates would be nucleated in the case of prolonged annealing times [14,15,16]. Increased configurational entropy in same cases may stabilize single-phase solid solution microstructures [2,4]. Generally, however, this effect is insufficient to override the driving forces that favor the formation of secondary phases by precipitation or decomposition. Thus, high configurational entropy cannot be considered a useful a priori predictor of whether a high-entropy alloy will form thermodynamically stable single-phase solid solutions or not [17]. 

For a CrCoMnFeNi alloy to be used in industrial applications, it is critical to investigate both the behavior of this HEA during prolonged use at high temperature and the manufacturing process, not to mention an accurate investigation into the microstructural evolution. In fact, precipitates could lead to relevant changes in the alloy’s mechanical properties, either increasing the strength or embrittling the alloy. In both cases it is important to control the new phase precipitation or dissolution upon heat treatment. Moreover, the kinetics and the mechanisms of a second phase precipitation into FCC HEAs are still poorly understood and this spurred us to further investigate them. Furthermore, it is proved that the secondary phases may contribute significantly to the HEAs’ properties, and it was reported that HEAs can overcome the strength-ductility trade-off when containing two or more phases [17,18]. 

With the aim to obtain a better understanding of the various factors that affect phase stability and therefore mechanical properties in high-entropy alloys, we undertook the present investigation into CoCrFeMnNi to determine what happens when the starting, mechanically pre-alloyed powders, are melted by SLM (selective laser melting) in additive manufacturing [19,20]. 

This processing method has the advantage of overcoming several limitations of the traditional processing methods. For instance, the ability to net-shape manufacture specimens with a high geometrical complexity without the use of dies. SLM was chosen because of its flexible layer-by-layer increase control feature. SLM does not require a vacuum environment and its equipment is therefore simpler than that of selective electron beam melting under vacuum (SEBM). It provides cooling rates up to 10^5^ K/s, while those of conventional melting processes (casting or welding) are typically less than 100 K/s. Therefore, SLM yields finer grains, a fine cellular dendrite structure and substructures within the grain, which improves the overall mechanical performance of the final components [21]. In fact, the hardness and tensile properties, at low and high temperature, are superior compared to alloys obtained with traditional casting and recrystallization [22,23,24]. 

The effect of non-equilibrium processing of SLM is also a great amount of residual stresses, a dense dislocation network, impurities and fine precipitation phases at boundaries [25]. This defective state was considered in this work as responsible for both the mechanical properties and microstructure evolution during high-temperature aging. Specifically, we reported on the tetragonal σ phase formation in FCC CoCrFeMnNi after thermomechanical processing comprising a room temperature thickness reduction of 90% and subsequent prolonged annealing. Prolonged annealing times were used because of the sluggish diffusion effect that slows down phase transformation. In addition, cold working was used to accelerate phase precipitation [26,27]. 

Although it is possible to use them as structural materials in high-temperature environments, studies on equiatomic CoCrFeMnNi HEAs produced by additive manufacturing have been limited to manufacturing, microstructural analysis and room temperature tensile properties [28,29,30,31,32]. There have been no studies that investigated the effect of the unique microstructure of SLM-built HEAs on the high-temperature mechanical properties and deformation mechanism. Consequently, the objective of this work was to document the fine secondary σ phase precipitation in the as-built state induced by SLM technology. Being present from the beginning, a prolonged annealing was not required in order to see it appear, as is the case for HEAs produced with the traditional melting process [13,15,16]. In particular, this phase, together with a further fine nitride precipitation due to the nitrogen protective atmosphere, which is commonly used in industry in order to limit oxidation, was the unavoidable consequence of the thermal cycles that accompanied the layer by layer growth of the alloy. There is abundant literature regarding the production of steel and other non-ferrous alloys with SLM technology in nitrogen atmosphere. Not that much exists for the case of Cantor’s alloys [33,34,35,36]. In particular, there is a lack of a precise correlation between the microstructural evolution and the associated mechanical properties. Unlike the well-known effect due to a protective nitrogen atmosphere, that is, nitride precipitation, here, we observed an associated further precipitation which occurs simultaneously. 

Whereas up to a certain working temperature fine precipitates can play a reinforcing role, starting from a temperature of about 950 K, the σ phase tends to swell and to preferentially place itself on the grain boundaries of the alloy. This in turn can lead to a significant reduction of the mechanical and chemical properties. This is described in the following and confirmed by a mechanical characterization with microhardness and hardness measurements that correlate with the microstructural evolution that occurs during aging. 

## 2. Materials and Methods

For this study, a nominally equiatomic HEA was produced by an SLM process. Co, Cr, Fe, Mn and Ni powders, with purity greater than 97% at. were supplied by Sigma Aldrich (Darmstadt, Germany). The powders were mechanically alloyed in an inert Argon atmosphere, using a Retsch (Haan, Germany) PM 100 high energy planetary ball mill and steel balls. Treatment cycles lasting 15 min each with a 5 min break between them for a total grinding time of 45 h were used [37]. Break time was used to avoid overheating. 

The SLM apparatus, used to produce samples with volume 50 × 5 × 5 mm^3^, was a SISMA MYSINT100 RM (Vicenza, Italy). A high purity Nitrogen atmosphere was used in order to minimize oxidation during the production process. Melting did not take place until the oxygen level dropped below the set limit threshold, which was 0.5%. The machine process parameters were H = 0.05 mm, F = 200 J/mm^3^, recoating rate= 70/150 mm/s with a constant deposition thickness and spot diameter of 20 μm and 50 μm, respectively. The samples’ surface, perpendicular to the growth direction, was divided into 6 slices filled according to a chessboard strategy [38].

Samples in the: (1) as-built state, (2) cold-rolled state, (3) after annealing at 1170 K for 100 h and (4) after thermomechanical treatments in the temperature range 723–1423 K for 25 h, were sectioned to the deposition direction and were prepared by standard grinding and polishing methods. This included polishing with diamond suspension down to 1 μm when preparing for metallographic observations in optical and scanning microscope and finishing with colloidal silica solution with a particle size of 0.05 μm when preparing for EBSD characterization. 

The thermomechanical treatments, performed in order to speed up the formation and precipitation of intermetallic phases, consisted of cold rolling with a 90% thickness reduction followed by heating at one of the eight selected temperatures, that is: 723, 823, 923, 1023, 1123, 1223, 1323, 1423 K. Cold rolling was performed on sections of the as-cast disks using laboratory rolling equipment to achieve a thickness reduction from 3 mm to 0.3 mm. The displacement rate of the rolling mill was kept constant at 0.02 mm/s. The cold-rolled specimens were subsequently re-crystallized by a 25 h annealing in a Kanthal Super HT rapid high-temperature furnace (Hallstahammar, Sweden).

In all cases, subsequent phase characterization to identify the crystalline structures was carried out by X-ray diffraction (XRD) and by SEM-FEG-EDS-EBSD. XRD analyses were performed with an X’Pert PRO Panalytical diffractometer (Malvern, UK) equipped with a proportional gas detector in a Ɵ-2Ɵ configuration in the angular range from 10° to 120° using Cu Kα radiation (λ = 0.15406 nm). Microstructural investigations were carried out on cross sections of all samples by an optical microscope and by a field emission scanning electron microscope (SEM-FEG) Tescan MIRA3 (Brno, Czech Republic) equipped with an EDS (energy dispersive spectroscopy) microanalyzer Bruker (Billerica, MA, USA) Quantax. The calculated semi-quantitative EDS analyses, for each phase, were averaged out of five values.

In addition, crystallographic orientation and grain size were analyzed by electron backscattered diffraction (EBSD). EBSD maps were made with Quantax EBSD detector on as-built, cold-deformed, annealed and re-crystallized samples to document the FCC matrix alloy and the precipitation secondary phase. The EBSD data were recorded and analyzed using the Bruker Esprit software. Prior to any optical or electron observation, samples were polished and chemically etched with a thermal treatment at 473 K in a glyceregia solution composed of 1 HNO_3_ + 3 HCl + 3 Glycerol. As regards the mechanical properties, Vickers microhardness was evaluated in this study through indentation tests carried out with a VOLPERT tester (Zell am Main, Germany). Five measures were performed for each sample.

## 3. Results and Discussion

### 3.1. Mechanically Alloyed HEA

The morphology and particle size distribution of the mechanically alloyed powders are reported in Figure 1. The dimensional range distribution was obtained by means of a laser meter Mastersizer 2000 ver. 5.2 of Malvern Instruments (Malvern, UK). The corresponding X-ray diffraction, Figure 2a, mainly shows the reflections of a single FCC phase, together with a weak presence of a secondary phase and iron oxides.

Oxides are probably due to surface oxidation, due to the high surface reactivity of these powders, particularly in the case of iron. Mechanical alloying does not bring about spherical dust formation, as obtained with the (much more expensive) gas atomization technology. As shown in Figure 1b, most powders fell within a range between 10 and 100 μm. By using a sieve, it would be possible in the future to optimize the particle size distribution up to an appropriate average particle size. A dimensional optimization of the powders would lead to a more uniform microstructure, with dimensionally homogeneous melt pools and increased mechanical properties because of the macro porosity reduction.

Figure 2b documents the FCC crystallographic homogeneity throughout the entire thickness of each sample obtained by SLM. XRD was repeatedly performed after thinning a sample by abrasion. Number 1 refers to XRD before abrasion and number 6 to the innermost section after five abrasions whose overall removed thickness is 0.5 mm.

### 3.2. Microstructure of Selective Laser-Melted Equiatomic CoCrFeMnNi High-Entropy Alloy as Built

The optical microstructures obtained on transversal sections of the as-built SLM samples are shown in Figure 3. The image shows the typical melt pools generated by the laser beam, whose depth also depended on the powder grain size non-uniformity. It is also possible to observe a layered morphology of thickness between 100 μm and 200 μm, depending on the thickness of the deposited powder bed.

Due to the locally intense power delivered to these powders, strong junctions were formed between the layers. As visible in Figure 4, the most common morphology observed in this alloy when produced by SLM is columnar grains aligned towards the build direction [39]. Columnar grain growth and grain orientation were induced by the rapid cooling conditions and from re-melting due to successive layer deposition. Inside the grains, very fine dendritic structures, which grow in the direction of the thermal gradient, were also produced. This is clear evidence that the morphology, size, direction and texture of samples are influenced by the process parameters. These in turn affect the mechanical properties of a component directly assembled in near net shape.

Figure 5a shows an SEM image on a plane perpendicular to the build direction and Figure 5b the corresponding EBSD map. Very small precipitation of an intermetallic phase identified as σ phase is visible (blue), in an FCC matrix, whereas XRD analysis on the same plane (Figure 2b), only identified an FCC structure. Based on the findings from EBSD, XRD and EDS analysis, reported in Table 1, the produced alloy was confirmed to basically have an FCC structure and a good equiatomic compositional homogeneity.

As already mentioned, there was a surface oxide contribution associated with the high powder reactivity. In fact, visible in the microstructures of the as-built samples, Figure 5, and samples aged up to 823 K, Figure 6 and Figure 7, the solid FCC solution exhibited a fine dispersion of randomly distributed oxides. Such oxides were previously observed in arc-melted CrCoMnFeNi by other authors [40]. Even finer precipitates cannot be seen below this annealing temperature, see Figure 8, since both σ phase and the other precipitates mostly nucleate and grow at temperatures between 900 K and 1300 K [41]. Based on a critical review of the literature, it is worth emphasizing that it is still not established whether, or which kind of, precipitates are present in the as-built CoCrFeMnNi alloy produced by SLM [42]. 

Fine precipitate formations at the edge of the microdendritic structures were documented in the BSE high resolution images of Figure 9 and the EDS microprobe analysis of Figure 10. Black precipitates were observed. They are nitrides, around which other precipitates with nanometric size grow. They are due to σ phase. This arrangement, with σ phase growing around nitrides, suggests that at first nitrides appear and then the σ phase is formed by heterogeneous nucleation. σ phase and other precipitates mostly nucleate and grow at temperatures between 850 K and 1250 K, as also observed by other authors [42]. This peculiar microstructure occurs because nitrogen interstitial diffusion during the SLM process, in the melted state, is fast and the subsequent rapid cooling does not allow the achievement of a stationary condition.

Unlike casting or other conventional manufacturing methods, materials produced by additive manufacturing can experience different thermal histories between layers. The variation in thermal history can in turn lead to different local microstructures, consequently affecting the precipitation process. Highly distorted areas favor chromium segregation and therefore ease the structure transformation from austenite to σ phase. The mechanism of the σ-phase formation depends on microstructure, chemical composition, temperature and time duration of the ageing treatment. In fact, the small size of the dendrites gives rise to a high grain boundary surface, which considerably speeds up the diffusion process even at room temperature [25,43].

As already stated, this alloy is a solid nitrogen supersaturated solution and, as a consequence of repeated thermal cycles, nitrides are nucleated, which, however, have extremely small dimensions due to the fast cooling rate, Figure 9 and Figure 11. These small precipitates are difficult to detect or to chemically characterize, e.g., to distinguish between Cr_2_N and CrN in EDS analyses.

### 3.3. Microstructure Evolution of Selective Laser-Melted Equiatomic CoCrFeMnNi High-Entropy Alloy after Cold-Rolling and Aging in the 723 K–1423 K Temperature Range

As reported in several recent studies, a second phase could appear in this alloy after annealing [13,27,43]. Thus, the microstructural evolution of this alloy when subjected to prolonged annealing times was studied. To better evaluate the microstructural evolution and the second phases precipitation, we followed two different paths. The first was annealing on as-built samples, the second was annealing after cold-rolling deformation, which speeds up the precipitation phenomena at the edge of new and smaller equiaxed grains by recrystallization. The alloy behavior was the same with regard to precipitate formation and composition, except that recrystallization occurred at a lower temperature in cold-rolled samples. An interesting question, at this point, is: will the second phase precipitation be favored in SLM over the alloy produced by casting? Will it be compositionally and structurally different or similar? Depending on that, the two alloys would exhibit different behaviors in a high temperature exercise. Figure 12, Figure 13 and Figure 14 show the alloy microstructure of cold-deformed samples, annealed at 1173 K for 100 h samples, cold-deformed and heated at 1173 K for 100 h samples, respectively.

The cross-sectional image of Figure 12a showed the typical elongated grain microstructure of a cold-rolled sample. There was no evidence of cracks or voids. The corresponding EBSD map of Figure 12b revealed an FCC matrix (red) and a very thin tetragonal σ phase (blue), not detected in XRD, Figure 2. The sample annealed for 100 h at 1173 K, Figure 13a, retained the same elongated microstructure except for the σ phase, whose grains increased in size. This was particularly evident in the EBSD map of Figure 13b where the same colors as before are used. Up to the recrystallization temperature, there was no particular difference in the microstructural morphology and secondary precipitation phases between cold-rolled and annealed samples.

The precipitation effect became more evident when recrystallization occurred, as shown in Figure 14. The original columnar grains became rounded and, due to increased diffusion rates, the σ phase grains grew in size while remaining preferentially located at the FCC grain boundaries. The diffraction spectrum continued to show the majority σ phase alone, Figure 15.

The semi-quantitative chemical analysis of the recrystallized FCC phase revealed how it was depleted of Cr and Mn, leaving unchanged, with respect to the as-produced alloy, the atomic composition of the other elements. In contrast, the precipitation phase was enriched with Cr and N while depleted in Fe, Co Ni and Mn, as can be seen from the EDS composition in Table 2 and Table 3. This was observed for both the samples of Figure 13 and Figure 14 [44,45]. XRD measures confirmed the presence in the samples of an FCC matrix and of a Cr-rich phase, Figure 8.

Let us consider now, in greater detail, what happens on cold-rolled samples as a function of the annealing temperature. The microstructural evolution due to annealing at 723, 823, 923, 1023, 1123, 1223, 1323 and 1423 K for 25 h, resulted in a quite complex process. Figure 6 and Figure 7 show the optical and scanning electron microscopy transversal section images of this process, respectively. Up to 923 K, the optical and scanning electron microscopy analysis revealed that the microstructure always consisted of an FCC solid solution with a chemical composition close to the starting equiatomic one, as reported in the semiquantitative EDS analyses of Table 2. The metallographic observations in optical and electron microscopy made on samples aged at 723, 823 and 923 K, Figure 6, confirmed the presence of a fine precipitation at the grain boundaries. As the SLM HEA had a predominant microstructure made of ultrafine grains with abundant grain boundaries, a Cr-N rich σ phase precipitation was expected. The small size of these precipitates, however, did not provide a sufficient scattering volume to allow this phase to be detected by XRD diffraction. 

Things started to change when the recrystallization process occurred, that is above 1023 K. Precipitates (σ phase) were now clearly observed. They were found to consist of a Cr-(Co, Mn, Fe, Ni N)-rich phase together with a significant increase in nitrogen, see Table 3. The structure was that of a body-centered tetragonal lattice, which embrittles the alloy.

Generally, this phase is believed to have an A_x_B_y_ formula, with x and y approximately equal [46,47]. Cr has been identified as the major forming element, and it is found with a concentration of ~50 at.%. Hence, σ particles can be described as an ideal solid solution in which Cr atoms are the solutes while the other elements constitute the matrix. However, from the EDS results alone, it was not possible to determine unambiguously whether other phases, such as μ, χ, Cr-rich ferrite (α′), carbides or Laves phases, sometimes observed in standard FCC HEAs, were present [48]. 

A significant precipitation was observed in the samples heat treated at 1023 K or above. It was a coarse σ-phase located at triple points of the recrystallized alloy. If the recrystallization was carried out at 1323 K for 25 h, there was also precipitation inside the FCC grains, close to the geminates, suggesting that these locations are preferential nucleation sites when the aging temperature increases, as reported in Figure 6 and Figure 7. The electron backscatter diffraction microstructures of Figure 14b show, as a significant example, the phase map of the recrystallized alloy at 1173 K which identifies a predominantly FCC alloy, red color, with geminates and a secondary sigma phase precipitation.

From a general point of view, precipitate sizes were found to increase with either increasing aging time or temperature. Therefore, it seems reasonable that the activation energy value of the σ-phase precipitation is the sum of three different contributions, i.e., those of lattice, pipe, and interphase boundary diffusion having high, intermediate, and low activation energies, respectively. The precipitation was the result of two phenomena (nucleation and growth) which were affected by the alloy’s microstructure. Therefore, further experimental work focusing on nucleation and coarsening is required to fully understand this process.

The X-ray diffraction patterns of the cold-rolled and annealed samples below the recrystallization temperature (up to 923 K) remained the same, as shown in Figure 8. Diffraction peaks from the matrix due to a single FCC phase were observed in all the samples up to 1423 K, but when the annealing temperature reached 1023 K, low-intensity diffraction peaks belonging to the σ phase could also be detected. The appearance of low-intensity diffraction peaks due to the σ phase was also reported by other authors [49]. 

The heterogeneous nature of nucleation significantly increases the rate of nucleation for temperatures close to equilibrium. Consequently, the growth of nuclei implies two distinct phenomena: (1) atoms of both phases jump from one side of the interface to the other and (2) atom diffusion from the solid solution towards the precipitate. Depending on which process is prevalent, the precipitation process will be controlled by interface phenomena or by diffusion. As suggested by our observations of temperature-dependent precipitation kinetics, which is consistent with a diffusion mechanism, it seems that the precipitation growth might be the result of the shift from one mechanism to the other, with diffusion becoming the prevalent one. In fact, the EDS analyses of the high-temperature annealed samples revealed a significant Cr depletion of the FCC structure, together with a Cr enrichment in the precipitates. Cr depletion in the FCC phase also correlates with a significant increase in the alloy grain size with the aging temperature, see Figure 5 and Table 2.

Precipitates observed after annealing at a temperature of 1323 K bear some resemblance to dendritic growth; they consist of plates or needles which lie on certain crystallographic planes of the matrix and preferentially grow in length by slightly modifying their transverse dimensions, as shown in Figure 6 and Figure 7. This kind of precipitate, that occurs inside the enlarged grains, is consistent with the idea that diffusion at the grain boundaries and volume diffusion have similar speeds. When the grains are larger, Cr atoms must diffuse over longer distances in order to reach the grain boundary. 

This shape evolution in precipitates anticipates their dissolution occurring at 1423 K, as shown in the micrographs of Figure 6 and Figure 7 and in the corresponding XRD spectra, where the new coarse grains were almost completely devoid of precipitates. After annealing at 1423 K, the semi-quantitative composition provided by EDS confirmed that the alloy became substantially monophasic with FCC lattice and a chemical composition which was approximately that of the starting equiatomic alloy, with the usual exception of Mn.

In the electron backscattering image of Figure 7, the correlation between grain size and precipitate size was clearly visible: the smaller the grain, the larger the number of precipitates because grain boundary diffusion is favored. In contrast, when aging temperature was higher than 1323 K, a significant presence of precipitates was found between the FCC coarse grains. It is reasonable to think that volume diffusion at temperatures T ≥ 1323 K preferentially occurs along high-density lattice planes because these are the most widely spaced planes. 

In summary, fine precipitates were present in as-built samples. Their size gradually increased with annealing temperature up to 1323 K to subsequently decrease at higher annealing temperatures. Annealing temperature also affected the composition and the place of formation of the precipitates: at the triple points of the grain boundaries when the alloy recrystallized (1023 K or above), along the columnar grain boundaries with rounded shape below 1023 K. Moreover, the annealing time required to accomplish precipitation was greater than 25 h. 

### 3.4. Mechanical Behavior: Microhardness Measures

Figure 16 shows the microhardness of HEA SLM samples before and after prolonged annealing treatments in the 723–1423 K temperature range. The microhardness measures were rather accurate for the cold-rolled and 1423 K annealed samples, having a relative uncertainty of about 1%. The relative uncertainty was higher (4%) at lower temperatures, between 723 and 1323 and for as-built samples (6%).

Microhardness increased significantly with cold-rolling and after annealing, mostly at 723 K or 823 K. This was reasonable, due to the previously mentioned fine precipitation at the grain boundaries. At higher temperatures, annealing led to a significant change in the volume fraction of the σ phase, whose maximum was observed around 1200 K. 

At 923 K, a softening effect began, also documented by the Cr depletion in the FCC solid solution, as shown in Table 2, which caused a reduction in the microhardness value in comparison with that of specimens annealed at lower temperatures. It is worth reiterating that deformation twinning was reported in the literature for CoCrFeMnNi as a mechanism of room temperature deformation [50,51].

When annealed at 1023 K, the alloy recrystallized and this brought about a microhardness increase. Subsequent annealing at higher temperatures entailed an increase in both precipitates and FCC grains. This in turn led to a progressive microhardness reduction up to the complete dissolution of the σ phase into a single FCC solid solution. The result follows the classical Hall–Petch relationship.

Correlating with the microstructural and XRD investigations, these mechanical tests can provide important information on the stability and metastability of these high entropy alloys.

## 4. Conclusions

Unlike the classical Cantor alloys obtained by casting technology, the alloy obtained by SLM did not require prolonged exposure to high temperatures to produce secondary phase precipitation.SLM technology with Nitrogen protective atmosphere induced a fine secondary phases precipitation of nitrides and intermetallic phases in the as-built state.The cold-rolled plus annealed FCC matrix became unstable and secondary phases precipitated. The observed precipitation was made of tetragonal Cr-rich σ phase and of mixed Cr-(Fe, Mn, Co, Ni)-nitrides which nucleated at the triple points.The size of the σ phase precipitates increased with annealing temperature. When the annealing temperature was below the recrystallization temperature, the size of precipitates was small due to a low diffusion rate and confined at the grain boundaries of the as-built dendritic structure.When the temperature exceeded 1323 K, the σ phase dissolved and the alloy substantially returned to single phase with near equiatomic composition.σ phase and nitride precipitations yielded a significant alloy hardening only at room temperature and for annealing up to 923 K, that is, below recrystallization temperature.Recrystallization, favored by cold rolling, induced, in the 1023 K–1323 K temperature range, the precipitation of a chromium- and nitrogen-rich phase. This effect led to a progressive hardness reduction of the alloy as a function of annealing temperature which can be understood recalling that Cr is a strong hardener in solid solutions, together with grain swelling.These experimental results suggest that thermodynamic databases and models need to be extended and improved to better understand this complex kind of alloy. The observed phase instability of the CoCrFeMnNi HEA suggests that care needs to be given to the alloy in the exercise of temperature.

## Figures and Tables

**Figure 1 materials-15-05544-f001:**
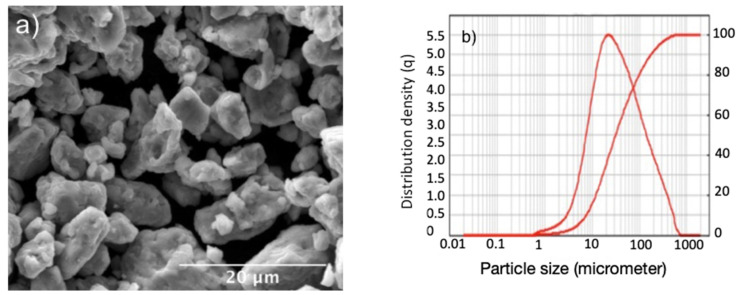
(**a**) Morphology of the mechanically alloyed powders; (**b**) particle size normalized and cumulative distribution.

**Figure 2 materials-15-05544-f002:**
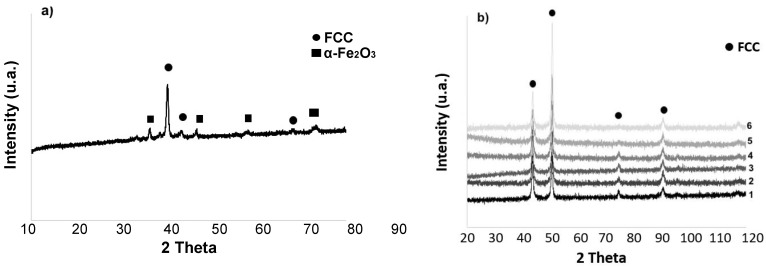
(**a**) XRD of mechanically alloyed powders; (**b**) XRD of the HEA after layer by layer abrasion (from 1 to 6). Overall removed thickness is 0.5 mm. Peaks are due to FCC Cantor’s alloy (card number 00-033-0397), Fe_2_O_3_ (card number 01-085-0599).

**Figure 3 materials-15-05544-f003:**
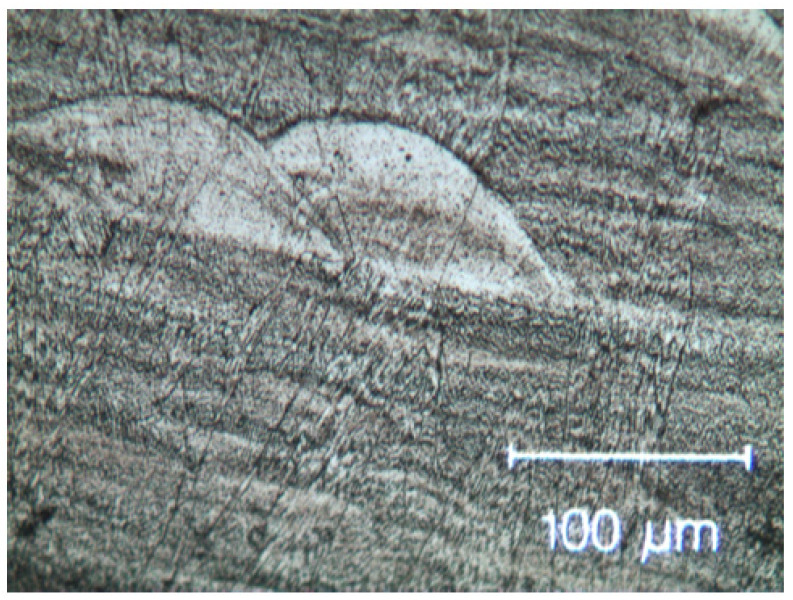
Optical micrographs of the as-built SLM HEA observed in transversal section with melt pots due to the laser beam.

**Figure 4 materials-15-05544-f004:**
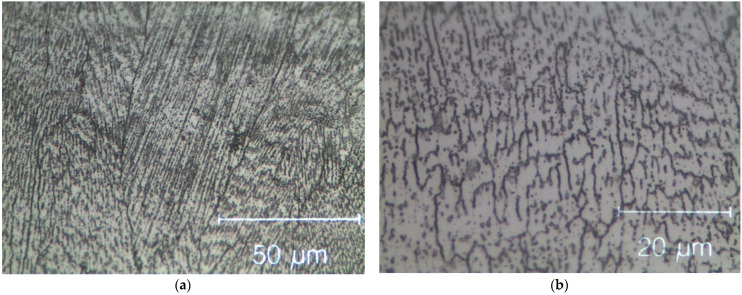
Optical micrographs of the as-built SLM HEA observed in transversal section. (**a**) Columnar grains aligned towards the build direction. (**b**) Dendritic structures inside the grains.

**Figure 5 materials-15-05544-f005:**
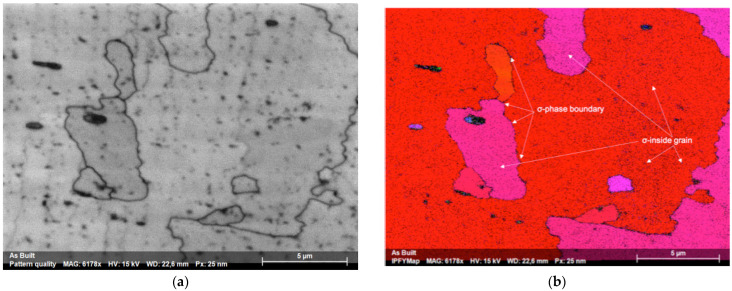
(**a**) Cross-sectional image of the as-built alloy. (**b**) corresponding EBSD map, with FCC matrix in red-pink color and σ phase precipitates in blue color.

**Figure 6 materials-15-05544-f006:**
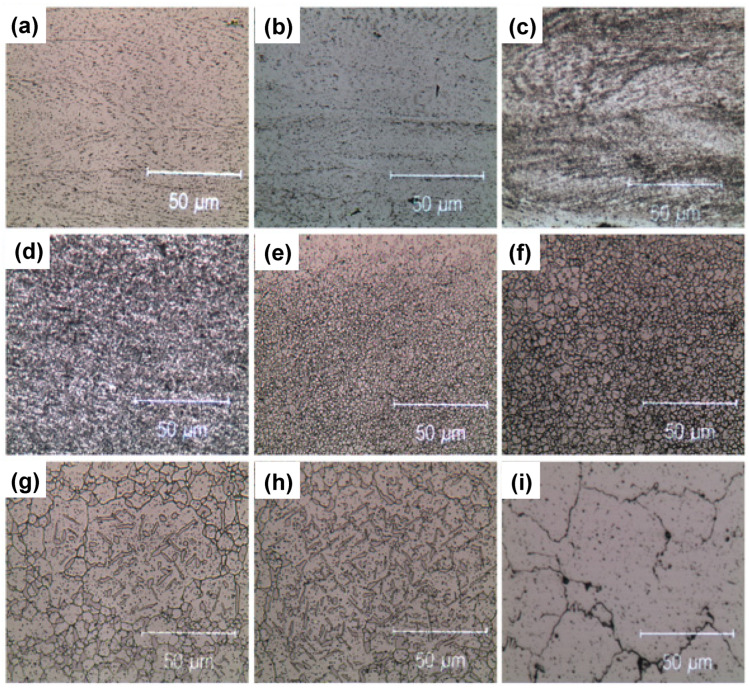
Optical transversal sections of cold-rolled (**a**) and cold-rolled plus annealed samples (from **b**–**i**). Annealing temperatures are: 723 K (**b**), 823 K (**c**), 923 K (**d**), 1023 K (**e**), 1103 K (**f**), 1223 K (**g**), 1323 K (**h**), 1423 K (**i**).

**Figure 7 materials-15-05544-f007:**
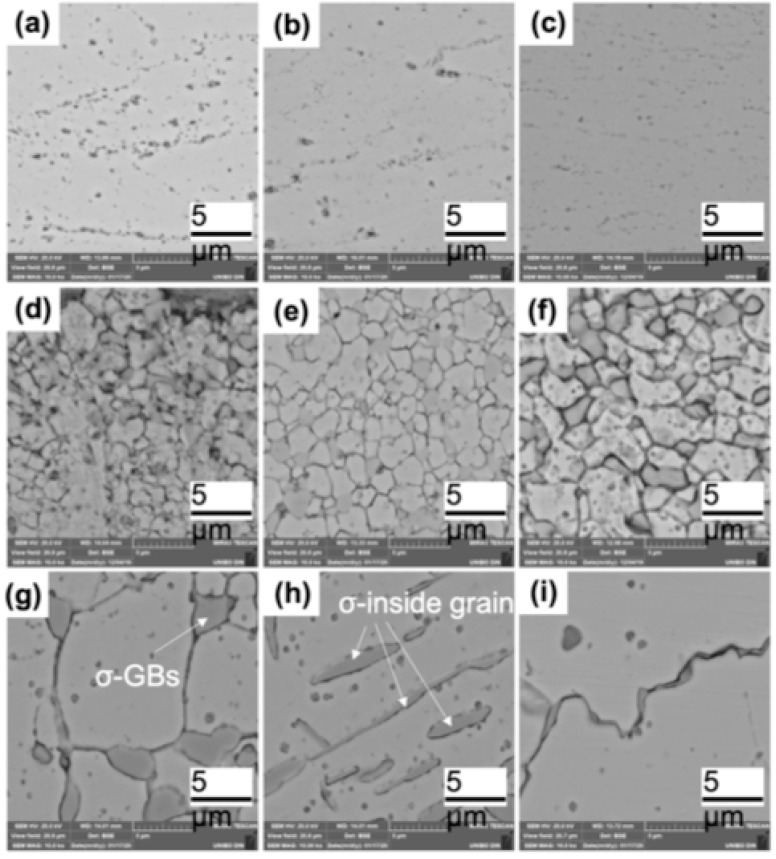
SEM-FEG transversal section microstructures of cold-rolled (**a**) and cold-rolled plus annealed samples (from (**b**–**i**)). Annealing temperatures are: 723 K (**b**), 823 K (**c**), 923 K (**d**), 1023 K (**e**), 1103 K (**f**), 1223 K (**g**), 1323 K (**h**), 1423 K (**i**).

**Figure 8 materials-15-05544-f008:**
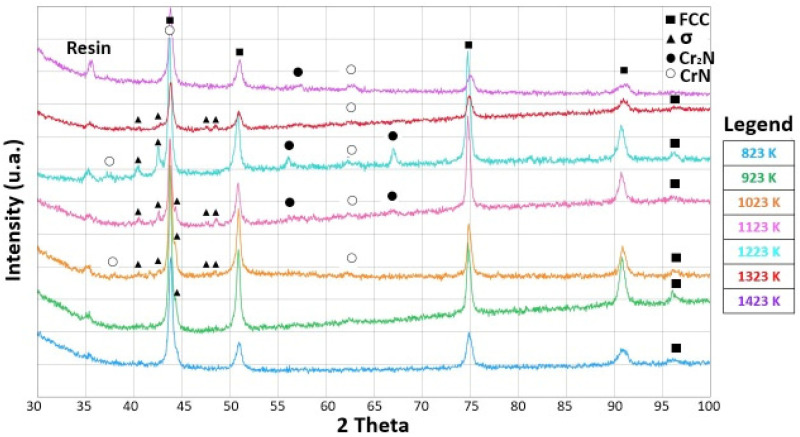
XRDs of cold-rolled and annealed samples at temperatures between 823 K and 1423 K. Temperature is increasing in 100 K steps, bottom up. The sequence is: 823 K (blue), 923 K (green), 1023 K (orange), 1123 K (pink), 1223 K (light blue), 1323 K (red), 1423 K (magenta). Peaks are due to resin, FCC Cantor’s alloy (card number 00-033-0397), σ phase (card number 00-005-0708), Cr_2_N (card number 00-035-0803), CrN (card number 00-011-0065).

**Figure 9 materials-15-05544-f009:**
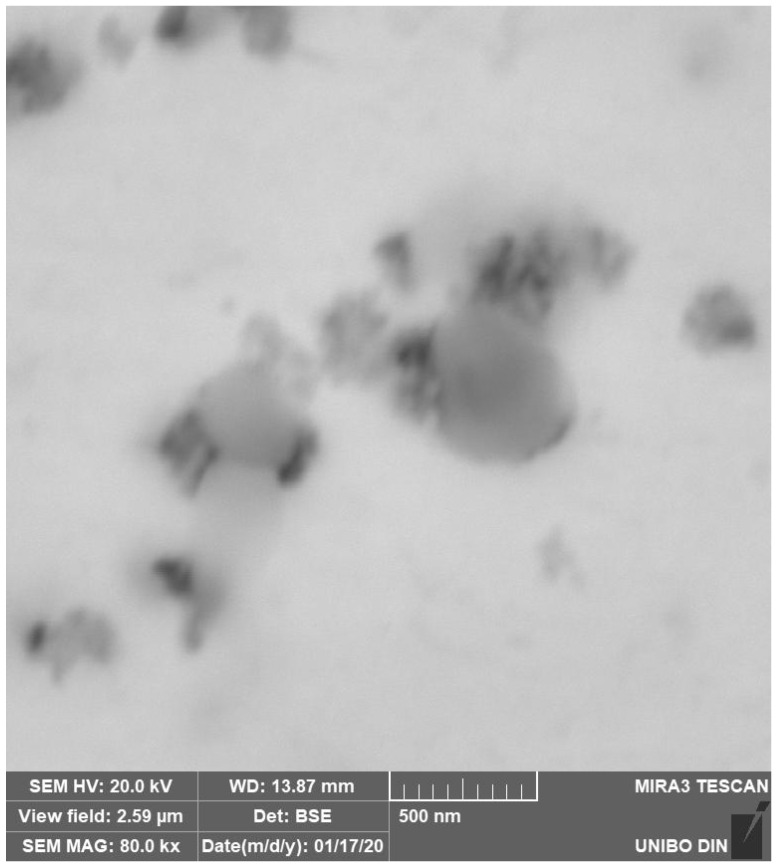
SEM FEG high resolution precipitate image in an as-built sample.

**Figure 10 materials-15-05544-f010:**
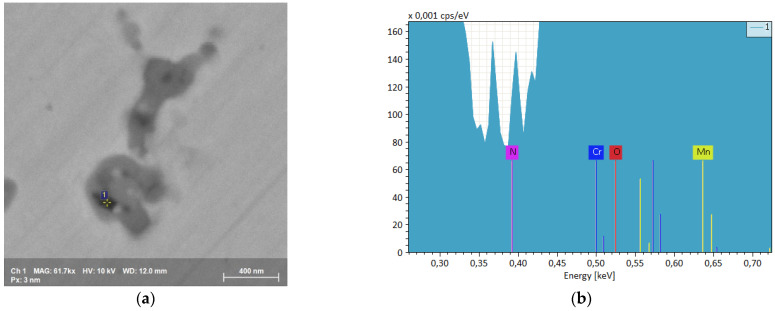
(**a**) SEM FEG high resolution precipitate image in an as-built sample. (**b**) EBSD spectrum of the black area of (**a**) containing a fine nitride precipitate.

**Figure 11 materials-15-05544-f011:**
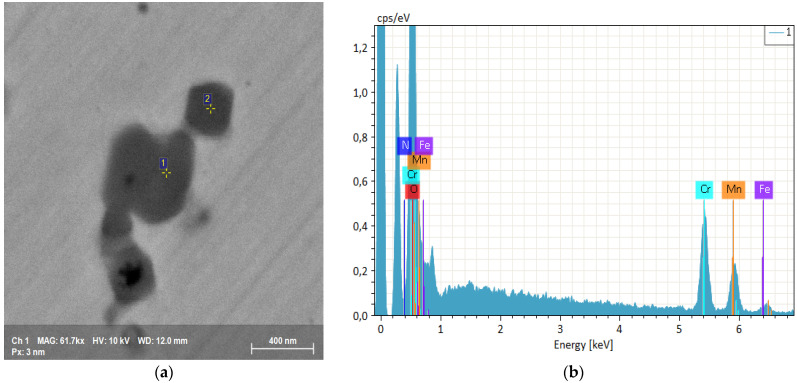
(**a**) SEM FEG high resolution precipitate image. (**b**) EBSD spectrum of the rounded σ phase.

**Figure 12 materials-15-05544-f012:**
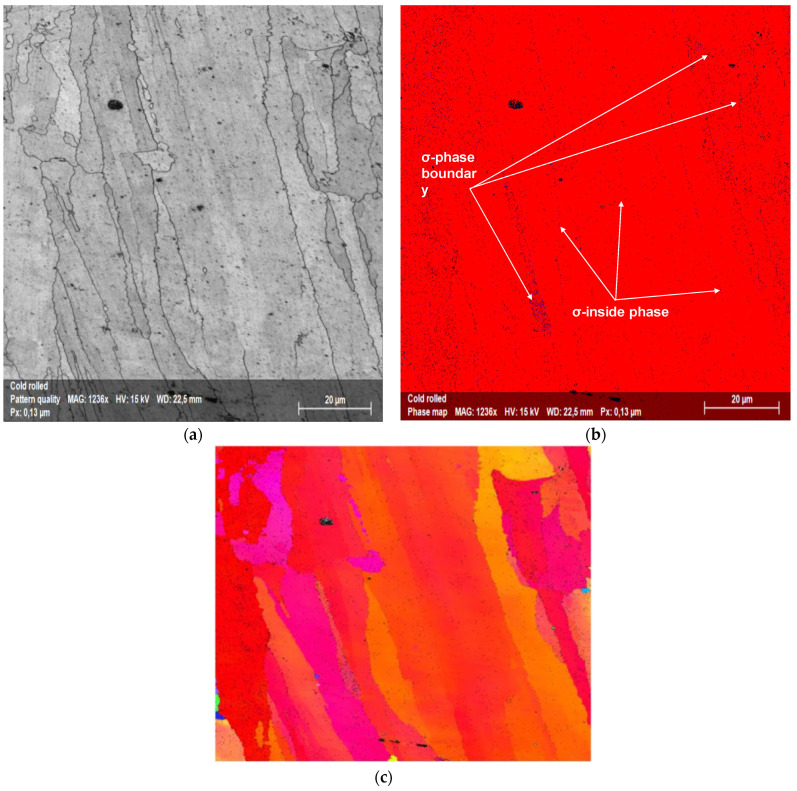
(**a**) Cross-sectional image of a cold-rolled sample. (**b**) corresponding EBSD map, with FCC matrix in red and σ phase precipitates in blue. (**c**) Inverse pole figure along *Y* direction, IPFY, of the same sample. (**d**) IPFY color scheme.

**Figure 13 materials-15-05544-f013:**
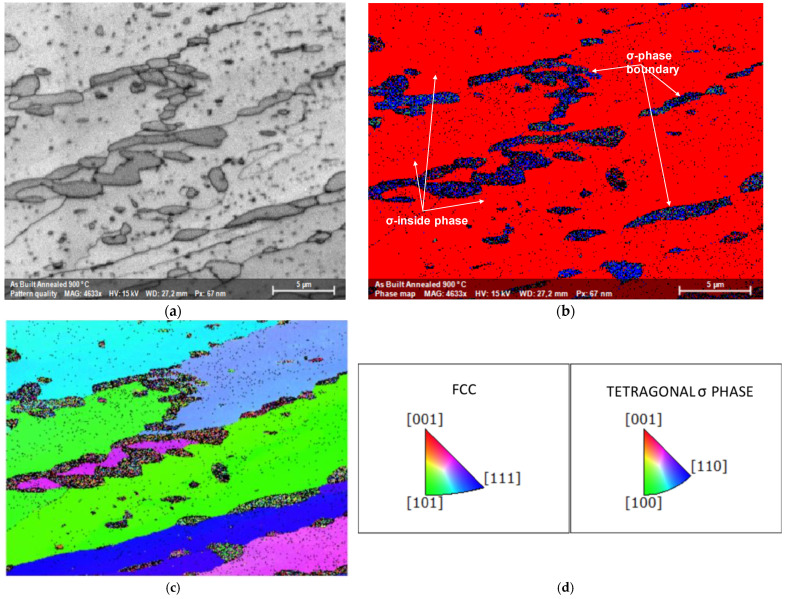
(**a**) Cross-sectional image of an as-built sample annealed at 1173 K for 100 h; (**b**) corresponding EBSD map, with FCC matrix in red and σ phase precipitates in blue. (**c**) Inverse pole figure along *Y* direction, IPFY, of the same sample. (**d**) IPFY color scheme.

**Figure 14 materials-15-05544-f014:**
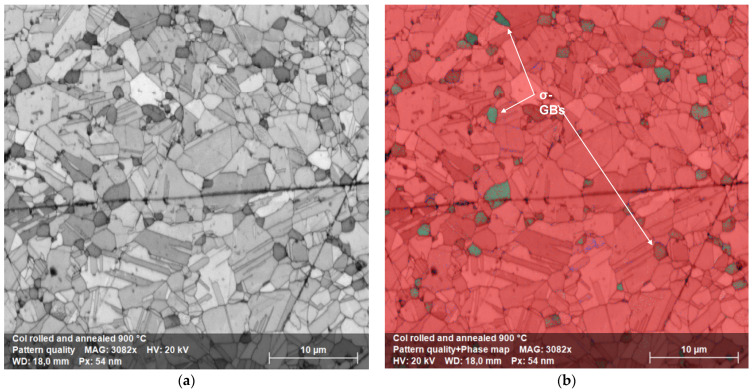
(**a**) Cross-sectional image of a sample cold-rolled and annealed for 100 h at 1173 K. (**b**) Corresponding EBSD map, with FCC matrix in red and precipitates in green.

**Figure 15 materials-15-05544-f015:**
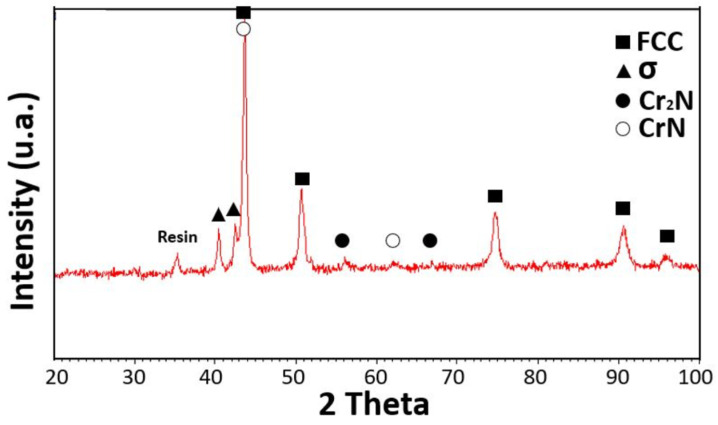
Diffraction pattern on a sample annealed at 1173 K for 100 h. Peaks are due to resin, FCC Cantor’s alloy (card number 00-033-0397), σ phase (card number 00-005-0708), Cr_2_N (card number 00-035-0803), CrN (card number 00-011-0065).

**Figure 16 materials-15-05544-f016:**
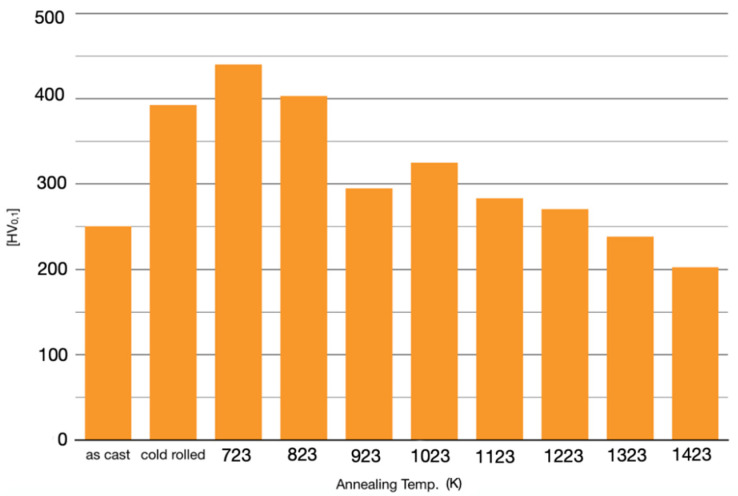
Microhardness of cold-rolled and annealed samples at temperatures from 723 K to 1423 K.

**Table 1 materials-15-05544-t001:** Semiquantitative EDS chemical analysis (atomic %) of as-built alloy. Relative uncertainty on data is less than 2% except for N (30%) and Mn (7%).

Element:	Nitrogen	Chromium	Manganese	Iron	Cobalt	Nickel
Atomic composition (%):	1.2	20.3	19	20.1	19.4	20

**Table 2 materials-15-05544-t002:** EDS semi-quantitative chemical compositions, in atomic %, of FCC matrix of cold-rolled samples. Relative uncertainty on data is less than 2% except for N (30%) and Mn (7%).

Temperature (K)	Nitrogen	Chromium	Manganese	Iron	Cobalt	Nickel
As cast	1.2	19.8	14.4	21.7	22	20.9
723	0	19.7	16.3	22.5	21.7	19.8
823	1	18.9	17.5	20.5	21	21
923	0.6	18.2	15.3	22	22.4	21.5
1023	3	15.6	16.6	21.8	22	21
1123	1.1	14.2	16	23	23.4	22.3
1223	1.7	10.9	16.4	23.7	24.3	23
1323	1.3	10.8	17	24.1	24.1	22.7
1423	1.7	18.2	14	21.3	22.6	22.2

**Table 3 materials-15-05544-t003:** EDS semi-quantitative chemical compositions, in atomic %, of sigma phase of cold-rolled samples. Relative uncertainty on data is less than 2% except for N (30%) and Mn (7%).

Temperature (K)	Nitrogen	Chromium	Manganese	Iron	Cobalt	Nickel
923	1.4	19	16	21	21.8	20.8
1023	2.9	27.1	14.7	19	18.6	17.7
1123	9	39.3	10.7	14.9	13.7	12.4
1223	17	56.9	8.2	7	5.9	5
1323	22.5	45.1	8.7	9	7.8	6.9
1423	6.1	24	13	19	19.5	18.4

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
