# Peer review of "Microstructural Study of CrNiCoFeMn High Entropy Alloy Obtained by Selective Laser Melting"

_materials, 2022, doi:10.3390/ma15165544_

Round 1

Reviewer 1 Report

1. The problem of "abbreviation": if one term, like sigma phase, HEA, has been nominated, please use them consistently.

2. Introduction part: some contents in relation to the experimental design, please generally organize it. More details should be described in the Materials and Methods.

3. The problem of tense: Most of the full text was written in the present tense, especially in the Materials and Methods, and also some parts in the Introduction and Results & Discussion. Please recheck the full text and revise it: the introduction of previous works, experimental procedure, and the results should be finished in the past tense.

4. XRD data of Figs. 2, 8, and 15, some obvious peaks were just ignored. Please carefully classify all peaks by referring to the PDF cards.

5. Fig. 2a: F2O3??

6. EBSD maps: please specify the map type, IPF? Some legends showing the crystallographic directions and phases cannot be absent. Also, please insert some necessary arrows to indicate the sigma phase when talking about its position, e.g. inside the grains or at the GBs.

7. SEM-EDS analysis: please keep in mind that when the region of interest is smaller than 1 um, the precision would sharply decrease. For such cases, TEM- or STEM-EDS would be strongly recommended for use. Accordingly, in Tables 1 and 2, the error bar should be inserted for each element at the least.

8. In Figure 16, the error bar should be employed for the HV value of each case.

9. Conclusions: this part is not a "Discussion" part. Please pinpoint the core findings or the main output in a more concise way.

Author Response

Dear reviewer, in the attached file You will find the response to your comments.

Sincerely Yours,

Enrico G. Campari

Reviewer 2 Report

It is very interesting to note that atmospheric nitrogen is contained within the alloy sample. Hence, more objective evidence is needed. For example,

1. Can you provide EDS mapping of the cross section? The point analysis in Figs. 10 and 11 is not sufficient.

2. Can you show another paper that reports nitrogen deposition in SLM, and add some discussion?

3. Have you ever fabricated your HEA sample under argon atmosphere?

In addition, minor comments;

- Please proofread your manuscript and check English again. (e.g. "Owing due to that," in line 103)

What does EBDS stand for, and is it different from EBSD?

Author Response

(The authors gave the same response as above.)

Reviewer 3 Report

The article is devoted to a popular topic among researchers, namely, high-entropy alloys. In the present work, material samples were obtained by selective laser melting.

I had questions and comments when reviewing:

1. For what purpose were the samples obtained by the SLM method subjected to cold rolling? What is the degree of deformation? What equipment was used for cold rolling? All this should be described in 2. Materials and Methods.

2. Line 133 refers to the use of thermomechanical processing. It is necessary to give a description of this processing and explanatory graphics modes. All this should be described in 2. Materials and Methods.

3. Lines 132...134 say "Sample in the: (1) as built state, (2) cold rolled state, (3) after annealing at 1170 K for 100 h and (4) after thermomechanical treatments into the temperature range 723-1423 K

in 100 K steps for 25 h". As I understand there are only 4 samples, and in Figures 6 and 7 there are nine samples. So how many samples?

4. The photo in figure 6 does not have signatures (a), (b) .... (f).... It will be very difficult for the reader to understand which mode is which, he will have to count the symbols of the drawings with the modes each time: 723 K, 823 K, 923 K, 1023 K, 1103 K, 1223 K, 1323 K, 1423 K.

5. The photos in Figures 3, 9,10 and 11 are not in focus. They need to be replaced with better ones.

Author Response

(The authors gave the same response as above.)

Round 2

Reviewer 1 Report

Several points should be carefully improved before publication.

1. please add the PDF-# to each peak of the XRD figures.

2. In the IPFY maps, IPF triangles are absent.

3. Regarding the HV values, error bars or relative uncertainty cannot be found in Fig. 16 or section 3.4.

4. Despite being divided into several items, conclusions are too long-winded. Please modify it more precise and clean. Reference appearing in the Conclusions is not accepted.

Author Response

Dear referees,

we are submitting a revised version of the article:

Microstructural study of CrNiCoFeMn high entropy alloy obtained by selective laser melting

By: E.G. Campari , A. Casagrande

In the following it is reported a detailed list of all changes to the previous version of the work done according with your suggestions and comments. In the article, these changes are reported in red.

Referee 1

Comments and Suggestions for Authors

Several points should be carefully improved before publication.

  1. please add the PDF-# to each peak of the XRD figures.

Card numbers are now added in the relative figure captions.

  1. In the IPFY maps, IPF triangles are absent.

IPF triangles are now added (figure 12d and 13d)

  1. Regarding the HV values, error bars or relative uncertainty cannot be found in Fig. 16 or section 3.4.

The relative uncertainty regarding HV values are now present in section 3.4.

  1. Despite being divided into several items, conclusions are too long-winded. Please modify it more precise and clean. Reference appearing in the Conclusions is not accepted.

Conclusions were shortened in order to make them more clear and precise. The Reference in the conclusions has been removed.

Enrico Campari

Reviewer 3 Report

The authors used my comments to improve the article. I recommend the article for publication in this version.

Author Response

(The authors gave the same response as above.)
